# Does Cranial Base Angle Make a Difference in the Effectiveness of Functional Orthopedic Treatment? A Retrospective Cohort Study

**DOI:** 10.3390/jcm14010096

**Published:** 2024-12-27

**Authors:** Taner Öztürk, Uğur Topsakal, Gulsumkhanım Vahabova, Ahmet Yağcı, Eldar Sheydayev

**Affiliations:** 1Department of Orthodontics, Faculty of Dentistry, Erciyes University, Kayseri 38039, Türkiye; dtahmetyagci@hotmail.com; 2Institute of Health Sciences, Erciyes University, Kayseri 38039, Türkiye; ugurtopsakal@hotmail.com (U.T.); vahabovagulsum@gmail.com (G.V.); seydayeveldar@gmail.com (E.S.)

**Keywords:** functional orthopedic therapy, cranial base angle, cephalometric radiography, mandibular retrognathia, incisor parameters, vertical facial dimensions

## Abstract

**Background/Objectives:** The literature suggests that the cranial base angle is considered one of the contributing factors to sagittal jaw malpositions when its relationship with the viscerocranium is examined. Our study aims to compare and evaluate the outcomes of fixed functional orthopedic treatment in patients with mandibular retrognathia across different cranial base groups. **Methods:** Participants were treated at Erciyes University with fixed functional appliances and categorized by CBA into low (<130°), medium (130°–134°), and high (>134°) groups. A total of 39 patients were included: 13 in the low CBA group (7 males, 6 females; mean age 14.62 ± 1.12 years), 13 in the medium CBA group (3 males, 9 females; mean age 14.38 ± 0.96 years), and 13 in the high CBA group (4 males, 9 females; mean age 14.08 ± 1.04 years). **Results:** In the low CBA group, Ar-Go-N (*p* = 0.005) and SNA (*p* = 0.023) angles significantly decreased, while the ANB angle and Wits appraisal significantly decreased across all groups (*p* < 0.05). The high CBA group showed increases in ANS-Me, N-Me, N-ANS, and N-Gn lengths (*p* < 0.05). The medium and high CBA groups had significant increases in S-Go and ANS-Gn lengths, while Co-Gn length increased significantly in the low and high CBA groups (*p* < 0.05). Incisor measurements (IMPA, L1-APog, L1/NB, L1-NB) increased in all groups, with overjet and overbite reduced (*p* < 0.05). U1/PP (*p* = 0.039), U1/SN (*p* = 0.043), U1-NA (*p* = 0.030), and U1/NA (*p* = 0.025) parameters increased in the low CBA group, with the Upper Lip–E distance decreasing significantly in the low and high CBA groups (*p* < 0.05). A comparison between groups showed significant differences in U1/PP, U1-NA, and U1/NA parameters, which increased in the low and medium CBA groups but decreased in the high CBA group. **Conclusions:** CBA influences treatment outcomes. The low CBA group experienced decreases in Ar-Go-N and SNA angles, while the high CBA group showed increases in certain vertical facial dimensions. Incisor parameters rose in the low and medium CBA groups but decreased in the high CBA group, suggesting limited CBA effects on treatment results.

## 1. Introduction

Class II malocclusion is among the most prevalent orthodontic problems, affecting approximately 33% of the population [1]. In nearly 80% of these cases, the condition is associated with mandibular retrusion [2]. Considering the anatomical relationship between the cranial base and the maxilla, as well as the role of the maxilla in mandibular rotation, investigating the impact of different cranial base angles on various orthopedic treatments for sagittal jaw malpositions holds significant importance [3].

Functional treatment is a therapeutic approach wherein forces occurring in the muscles and soft tissues surrounding the mouth during functional activities are transmitted to the dental and jaw structures through functional appliances, aiming to achieve changes in the sagittal and vertical alignment of the mandible [4]. In the literature, the first mention of functional treatment applications dates back to 1879, when Norman Kingsley introduced the prototype with his “Bite Jumping” appliance, a removable plate used to correct the position of the lower jaw [5]. Class II malocclusion, affecting about one-third of the population, is one of the most frequently encountered orthodontic issues [1]. The effectiveness of functional orthopedic treatments for these malocclusions is a widely discussed issue, with conflicting results in orthodontic literature [6,7]. Although there are different types of fixed and removable functional appliances used today, such as the activator [8], Bionator [9], Frankel [10], Twin Block [11], Herbst [12], and Forsus [13], they all utilize similar mechanisms to position the mandible forward. Considering the disadvantages, such as the importance of patient compliance, in removable appliances, patients using fixed functional appliances were included in our study to ensure more reliable results [14].

Various factors such as head and neck posture, basicranial morphology, and soft tissue tension are believed to affect the development of skeletal malocclusions. The contribution of cranial base angulation on the cause of sagittal jaw discrepancies remains a topic of ongoing debate [3]. The cranial base angle, also known as the saddle angle, is typically measured on radiographs between the sella, nasion, and basion points. However, to define the posterior boundary, the articulare and Bolton points are sometimes used, which can complicate the comparison between distinct studies. The angle is around 142 degrees at birth, decreasing to 130 degrees by the age of five. It remains relatively stable between the ages of 5 and 15 [15,16]. Malta et al. reported that the anterior cranial base does not maintain a constant size and experiences growth throughout all stages of puberty (CS1 to CS6 in cervical maturation stages). They found that the distance of the anterior cranial base (measured from nasion to sella) continues to grow until early adulthood [17]. M. Afrand et al. [18] concluded that the anterior cranial base undergoes growth and development until adulthood, with the sella turcica remodeling backward and downward and the nasion moving forward due to the increase in size of the frontal sinus.

The longitudinal cephalometric study by A. Klocke et al. [19] investigated skeletal characteristics in individuals with reduced and increased cranial base angles. The ANB and N-A-Pg angles showed no significant differences. Individuals with a large cranial base angle exhibited a skeletal Class II tendency at both assessment points. Cephalometric variables at age 12 allowed the classification of 88.1% of individuals, indicating a stable skeletal pattern. The connection between skeletal jaw pattern and cranial base flexure is determined prior to the age of five.

Although numerous studies have evaluated the effectiveness of fixed functional orthopedic treatment, there is a lack of comprehensive analyses examining treatment outcomes in individuals with varying cranial base angles (CBAs). 

This retrospective cohort study seeks to examine how the cranial base angle (N-S-Ba°) impacts the results of fixed functional orthopedic treatment in individuals with mandibular retrognathia. The study categorizes participants into three groups based on their N-S-Ba° measurements—low, medium, and high—and examines how these groups differ in terms of treatment responses measured through cephalometric changes. Null hypothesis (H_0_): The treatment outcomes of fixed functional orthopedic treatment do not show any statistically significant difference among individuals categorized by different cranial base angles.

## 2. Materials and Methods

Designed as a retrospective cohort study, this research received approval from the Erciyes University Clinical Research Ethics Committee (Approval Date: 26 May 2021, Approval No: 2021/371). The study participants were individuals who presented to the Erciyes University Department of Orthodontics with mandibular retrognathia and underwent fixed functional orthopedic treatment. Informed consent forms were obtained from each patient and their parents before the study.

The inclusion criteria for study participants were as follows:ANB° > 4 degrees;Bilateral molar and canine complete Class II molar and canine relationship in the permanent dentition;Overjet exceeding 6 mm;Normal or decreased SN/GoGn°;Craniomaxillofacial structures clearly visible on lateral cephalometric radiography (9).

The exclusion criteria for study participants were as follows:History of previous orthodontic treatment;History of congenital anomalies or trauma affecting the dentofacial region;Presence of medical conditions affecting bone structure.

This study included a total of 39 patients, categorized into three groups based on their cranial base angle (CBA): the low CBA group comprised 13 patients (7 males, 6 females; mean age 14.62 ± 1.12 years), the medium CBA group included 13 patients (3 males, 9 females; mean age 14.38 ± 0.96 years), and the high CBA group consisted of 13 patients (4 males, 9 females; mean age 14.08 ± 1.04 years).

The participants in the study were categorized into three groups based on the N-S-Ba°, which indicates the cranial base angle. In the study of Dhopatkar et al. [15], the N-S-Ba° for Class II division 1 individuals was given as 134.4 ± 4.95°. Considering this value, those with values < 130° were determined as low cranial base angle, those with values 130° ≤ N-S-BA ≤ 134° as medium cranial base angle, and those >134° as high cranial base angle.

### 2.1. Sample Size Calculation

Since there is no study in the literature regarding different groupings of the N-S-Ba°, a sample size estimate was made based on the preliminary data of 5 patients, who each received functional orthopedic treatment in groups separated according to the N-S-Ba°. The sample size for the study was established by including at least 9 participants per group, based on N-S-Ba° values from the preliminary study, with a 95% power, a 0.05 margin of error, and an effect size of d = 1.851, using G*Power software (Version 3.1.3; Franz Faul Universitat, Kiel, Germany) [20]. To account for potential sample loss and to enhance reliability, a total of 39 participants were included in the study, with 13 individuals in each group.

Despite the limited sample size, the scarcity of similar studies in the literature highlights the importance of this research in proposing ideas, making it a valuable contribution to the field by offering a general research perspective.

### 2.2. Orthopedic–Orthodontic Interventions

Participants included in the study received either Forsus or Herbst appliances as their fixed functional orthopedic appliances. At the beginning of the treatment, oral impressions were taken from the patient, and a Herbst appliance was fabricated by a technician on the resulting plaster models. The appliance was cemented intraorally using glass ionomer cement. In the study, a cast splint Herbst appliance (Herbst Type I, Dentaurum, Ispringen, Germany) was utilized [11] (Figure 1A). On the day the appliance was fitted with sagittal activation, the incisors were positioned in either an edge-to-edge relationship or a slight Class III molar relationship at that time. The use of the Herbst appliance lasted an average of 9 months. After a slight sagittal Class III molar relationship was achieved, the device was removed, and a fixed orthodontic appliance (Roth Prescription, 0.018″ Slot Mini Master Brackets, American Orthodontics, Sheboygan, WI, USA) was applied. The orthodontic treatment concluded once a Class I molar and canine relationship was achieved, along with ideal overjet and overbite values.

In individuals treated with the Forsus appliance (3M Unitek Corp., Monrovia, CA, USA) (Figure 1B), 0.017″ × 0.025″ stainless steel wires were applied to both the lower and upper jaw in one session, following the leveling and aligning of patients continuing with fixed orthodontic devices (Roth Prescription, 0.018″ Slot Mini Master Brackets, American Orthodontics, Sheboygan, WI, USA). Then the appliance was attached to the bands on the first molar teeth in the maxilla and to the arch wire located just distal to the mandibular canine teeth. The device remained in place until an edge-to-edge incisor relationship, like that in individuals treated with the Herbst appliance, was achieved; orthodontic treatment concluded once all completion conditions were met. After a right and left Class I molar and canine relationship was achieved and a good overjet–overbite relationship and interdigitation were achieved, the orthodontic devices were removed. During the retention period, a lingual retainer was used between the first premolars in the mandible, while a removable essix retainer was used in the maxilla.

The orthopedic treatment durations were recorded as 0.69 ± 0.08 years in the low CBA group, 0.70 ± 0.07 years in the medium CBA group, and 0.70 ± 0.11 years in the high CBA group. The total orthodontic treatment durations were calculated as 2.62 ± 1.29 years in the low CBA group, 2.41 ± 0.35 years in the medium CBA group, and 2.15 ± 0.56 years in the high CBA group. The treatment durations did not show significant differences between the groups.

### 2.3. Measurements

Lateral cephalometric radiographs evaluated in both applications were taken at the start of the treatment and upon its completion. Cephalometric measurements were performed in a similar manner to previous studies in which they were used (Figure 2 and Figure 3, Appendix A) [15,21].

### 2.4. Statistical Analysis

Statistical analyses were conducted using Jamovi open-source statistical analysis software (The Jamovi Project, 2022, Version 2.5, Sydney, Australia) [22]. Categorical data were examined using the Pearson Chi-Square test, while the Shapiro–Wilk and Levene tests assessed normality and homogeneity, respectively. For within-subject comparisons, parametric data were analyzed with the Paired Samples-t test, and non-parametric data with the Wilcoxon Signed-Rank test. Intergroup comparisons utilized One-Way ANOVA for parametric data (with the Tukey test for post hoc pairwise comparisons) and the Kruskal–Wallis test for non-parametric data (with the Mann–Whitney U test for post hoc pairwise comparisons). A statistical significance threshold was set at *p* < 0.05.

## 3. Results

The individuals included in the study did not differ in terms of cranial base angle groups and the relationship between CVM stage, hand–wrist stage, and functional appliance type. CVM Stage 4 was most frequent in the high CBA group (54.0%), and Stage 5 was most common in the medium group (77.0%). In addition, orthopedic treatment duration, total orthodontic and orthopedic treatment duration, age, cervical vertebral maturation stage, and hand-wrist stage did not differ between groups. The mean orthopedic treatment duration was approximately 0.70 years across all groups, with a combined orthopedic and orthodontic treatment duration averaging 2.4 years (*p* = 0.337). Mean ages were similar among the groups, with no significant differences (*p* = 0.461). Initial cephalometric values and values obtained because of treatment did not differ between genders and appliance types. Males were more prevalent in the low CBA group (54.0%), while females dominated the medium and high CBA groups (69.2% each) (Table 1).

A comparison of the initial cephalometric values of the individuals according to the cranial base angle groups is given in Table 2, and it was determined that all parameters except the N-S-Ba angle and N-S-Art angles did not differ between the groups at the beginning (*p* < 0.001). The N-S-Ba angle showed a progressive increase across the groups, with values of 128.01° ± 2.02° in the low CBA group, 134.64° ± 0.92° in the medium CBA group, and 139.31° ± 2.42° in the high CBA group. Similarly, the N-S-Art angle increased significantly, measuring 122.62° ± 3.34° in the low CBA group, 128.24° ± 3.78° in the medium CBA group, and 132.08° ± 4.22° in the high CBA group. For other parameters, including sagittal and vertical skeletal measurements, incisor measurements, and soft tissue measurements, no statistically significant differences were observed between the groups (*p* > 0.05). Key measurements such as SNA, SNB, ANB, and Wits values were similar across the groups, suggesting comparable baseline skeletal relationships. Additionally, incisor measurements such as U1/SN and IMPA, as well as soft tissue parameters like the nasolabial angle and Lip–E line distances, showed no significant variation among the groups (Table 2).

An intragroup comparison of the changes occurring because of orthopedic and orthodontic treatment for the cranial base angle groups is given in Table 3. When cranial base measurements were analyzed, the Ar-Go-N angle showed a significant decrease solely in the low CBA group (*p* = 0.005). When sagittal and vertical skeletal measurements were examined, the SNA angle decreased significantly only in the low CBA group. The ANB angle and Wits appraisal decreased significantly in all groups (*p* < 0.05). ANS, Me, N-Me, N-ANS and N-Gn lengths increased significantly only in the high CBA group (*p* < 0.05). While S-Go and ANS-Gn lengths increased significantly in both the medium and high CBA groups, they did not change significantly in the low CBA group (*p* < 0.05). While Co-Gn length increased significantly in the low and high CBA groups, it did not change significantly in the medium CBA group (*p* < 0.05). When the measurements of the incisors were examined, the parameters IMPA, L1-NB, L1-APog, and L1/NB increased significantly in all groups, while overjet and overbite decreased significantly. The interincisal angle increased significantly in the low CBA group and increased significantly in the medium CBA group, but did not show a significant change in the high CBA group. The U1/SN, U1-NA, U1/PP, and U1/NA parameters increased significantly only in the low CBA group. When soft tissue measurements were examined, Upper Lip–E distance decreased significantly in the low and high CBA groups (Figure 4).

A comparison of the changes occurring because of treatment between groups is given in Table 4. Accordingly, while the U1/PP, U1-NA, and U1/NA parameters increased in the low and medium CBA groups, they decreased in the high CBA group, and these changes are statistically significant (*p* < 0.05). For cranial base measurements, no significant differences were detected in changes for most parameters, although trends suggest a reduction in N-S-Ba and N-S-Art angles in the high CBA group compared to increases in the low CBA group (*p* > 0.05). Sagittal and vertical skeletal measurements, including SNA, SNB, ANB, and Wits appraisal, showed no significant differences among the groups (*p* > 0.05). Incisor measurements revealed distinct patterns of change, with U1/PP, U1-NA, and U1/NA parameters showing statistically significant differences (*p* = 0.029, *p* = 0.019, and *p* = 0.012, respectively). These parameters increased in the low and medium CBA groups but decreased in the high CBA group. Other incisor parameters, such as IMPA and L1/NB, displayed consistent changes across all groups without statistical significance. Soft tissue measurements, including the nasolabial angle and Lip–E Line distances, exhibited minimal changes and no significant differences between groups (*p* > 0.05) (Table 4).

## 4. Discussion

Although research has explored the impact of appliances used in the functional treatment of Class II patients on cranial base angle, there are limited studies that compare the effectiveness of functional treatment across patients with varying cranial angles. This study aimed to assess the impact of cranial base angle on the results of fixed functional orthopedic treatment in individuals with mandibular retrognathia. The methodology and findings of this study support the null hypothesis, indicating no statistically significant variation in the treatment outcomes of fixed functional orthopedic treatment among individuals classified by different cranial base angles.

Michael Courtney et al. [23] conducted a study to evaluate the changes in the maxilla and cranial base resulting from treatment with the Fränkel function regulator and Harvold activator. The study involved 42 children aged 10 to 13 who presented with Class II, division 1 malocclusions. Lateral cephalometric radiographs were obtained at the beginning of the study and after 18 months of treatment. The findings revealed that both appliances effectively reduced overjet by palatally tipping the length of the maxillary arch. However, the study noted that neither appliance significantly impacted the horizontal or vertical positioning of the maxillary molars. Additionally, it observed small but noteworthy changes in the cranial base angle, which were attributed to considerable variations observed at the basion in some participants. The appliances did not impact the maxilla’s position. In our study, we similarly found that functional orthopedic appliances induced significant changes in various cephalometric parameters across different cranial base angle groups. Specifically, the Ar-Go-N angle and SNA angle decreased significantly only in the low CBA group, while the ANB angle and Wits appraisal decreased significantly in all groups. Significant increases in ANS-Me, N-Me, N-ANS, and N-Gn lengths were observed only in the high CBA group. Conversely, Co-Gn length increased significantly in both the low and high CBA groups. Our results also demonstrated that U1/SN, U1/PP, U1-NA, and U1/NA parameters increased significantly only in the low CBA group, while interincisal angle changes varied among groups. These findings highlight the importance of considering cranial base angle in treatment planning, as it may influence the outcomes of functional orthopedic interventions. Moreover, while Michael Courtney et al. observed minor changes in cranial base angle with the Fränkel function regulator, our study identified significant changes in multiple craniofacial dimensions dependent on the initial cranial base angle, suggesting a more pronounced effect of appliance type and patient-specific cranial morphology on treatment outcomes.

Okano et al. [24] reported in their study that the Herbst appliance was effective as a treatment option for Class II malocclusion, supported by various studies and meta-analyses cited in the references, and that no positional or morphological changes occurred in the cranial base. Our study provided evidence supporting this literature by showing that the cranial base angle did not significantly affect the effectiveness of fixed functional orthopedic treatments in patients with mandibular retrognathia from a clinical perspective.

The study by Bacon et al. [25] highlighted significant variations in the cranial base among Class II subjects, suggesting that while cranial base configuration may influence post-normal occlusion, it is not solely decisive. Their findings indicated that only 62% of Class II individuals could be accurately reclassified based on cranial base variables, emphasizing the necessity of considering additional factors when evaluating and treating Class II malocclusions. This reinforces the need for a comprehensive analysis that includes cranial base assessment in understanding and addressing Class II malocclusions. Our study complements this by showing that variations in cranial base angle do not significantly impact the effectiveness of fixed functional orthopedic treatment, suggesting that treatment outcomes are influenced by a broader range of factors beyond cranial base configuration alone. The analysis of incisor measurements in our study reveals significant alterations across different cranial base angle (CBA) groups, highlighting the complex interaction between cranial base morphology and dentoalveolar responses to functional orthopedic treatment. The observed significant increases in L1-APog, L1-NB, IMPA, and L1/NB parameters across all CBA groups suggest a consistent proclination of the lower incisors, regardless of the cranial base angle. This uniformity may indicate that the functional forces applied by the appliances effectively induce mandibular advancement, resulting in similar dental compensations in terms of incisor positioning.

The study by Bhattacharya et al. [26] investigated the association between cranial base angle and maxillofacial morphology in an Indian cohort, using a cephalometric analysis. The study categorized individuals into three groups based on their cranial base angle (NSAr) values and examined various angular and linear parameters to assess their correlations. The findings indicated that the cranial base angle plays a crucial role in determining mandibular position. Both the mandibular plane angle and Y-axis are affected, with a flattening of the cranial base angle leading to a clockwise rotation of the mandible. The study revealed that as the cranial base angle decreases, the maxilla tends to protrude, and the SNA angle increases. In our investigation, the mean SNA values in the low CBA group were higher than those in the high CBA group, consistent with the results of the aforementioned study. However, following functional treatment, the SNA angle significantly decreased within the low CBA group, while a small but non-significant increase was observed in the high CBA group. This disparity can be attributed to the fact that, in the low CBA group, a positive change was noted in the upper incisors, whereas a negative change occurred in the high CBA group. As the cranial base angle increases, the protrusive effect on the upper incisors diminishes, and the retrusive effect becomes more pronounced. This implies a reduction in the forces transmitted to the maxilla, which, although not statistically significant, may account for the observed positive change in the SNA angle in the high CBA group compared to the other groups. Additionally, the findings from our study showed that the cranial base inclination has no definite effect on the effectiveness of functional orthopedic treatment.

A review of the literature reveals numerous studies examining the effects of fixed functional appliances on facial changes.

Meyer-Marcotty et al. previously evaluated the impact of the Herbst appliance on the soft tissues of patients with a Class II malocclusion using 3D stereophotogrammetry. They reported an increase in total facial height and convexity angle after Herbst appliance treatment, attributing this to the skeletal mass of the anterior facial height during Class II malocclusion treatment [27]. Similarly, Güler et al. concluded in their study that the Herbst appliance increased total facial height, convexity angle, and labiomental angle [28]. In our study, post treatment with fixed functional appliances, particularly in the high CBA group, an increase in lower facial height and total facial height was also observed.

De Almeida et al. reported that the improvement in facial profile associated with Herbst appliances was primarily attributed to changes in the upper lip, with lesser contributions from alterations in the lower lip and soft tissue of the chin [29]. Similarly, Flores-Mir et al., in their systematic review, supported this perspective by concluding that although fixed functional appliances induce statistically significant changes in soft tissue profiles, the clinical relevance of these changes may be limited due to their modest magnitude [30]. Similarly, in our study, the Upper Lip–E distance significantly decreased in the low and high CBA groups.

In a study conducted by Brandao et al., the effects of Twin Block, dentally anchored Herbst (HDA), and skeletally anchored Herbst (HSA) appliances on the soft tissue profile of Class II, division 1 malocclusion patients during the pubertal growth spurt were evaluated. Measurements taken before treatment and after 12 months of active treatment revealed statistically significant improvements in all groups, particularly in the lower lip, sulcus inferioris, facial soft tissue convexity, H angle, and soft tissue pogonion. However, no statistically significant differences were observed between the treatment protocols, leading to the conclusion that all three appliances had a similar positive impact on the soft tissue profile [31]. In our study, unlike the results of this study, no statistically significant changes were observed in the lower lip and soft tissue convexity angle following treatment with fixed functional appliances.

Today, the increasing aesthetic expectations of patients make AI-assisted facial beautification applications a significant reference point in their pursuit of treatments aimed at facial aesthetics. The study conducted by Tomasik et al. highlights the significance of this perspective. They suggested that AI-assisted enhanced images can guide clinicians in soft tissue-focused and personalized treatment planning while offering valuable insights into meeting patients’ aesthetic expectations [32]. Consequently, modern orthodontics is evolving to focus more on soft tissues, without disregarding the importance of hard tissues.There are several limitations in terms of the generalizability of the findings obtained from this study. Although this study was planned because of the power analysis, the sample size was relatively small. The study focused only on patients with Class II malocclusion and excluded other types of malocclusions, which may provide a broader understanding of the effect of the cranial base angle. Future studies with larger sample sizes and the inclusion of various malocclusion classes are recommended to confirm and expand these findings. Longitudinal studies are also needed to assess the long-term stability of the observed cephalometric changes across cranial base angle groups.

## 5. Conclusions

Significant differences in treatment outcomes were observed in the study groups: the low CBA group showed decreases in the Ar-Go-N and SNA angles, while the high CBA group showed increases in ANS-Me, N-Me, N-ANS, and N-Gn lengths. Sagittal and vertical skeletal measurements varied, with the ANB angle and Wits appraisal decreasing in all groups. Incisor parameters increased in the low and medium CBA groups but decreased in the high CBA group. In addition to these, we can say that cranial base inclination does not have a definite effect on the dental or skeletal effectiveness of functional orthopedic treatment. These findings underscore the impact of cranial base angle on orthodontic and orthopedic treatment results.

## Figures and Tables

**Figure 1 jcm-14-00096-f001:**
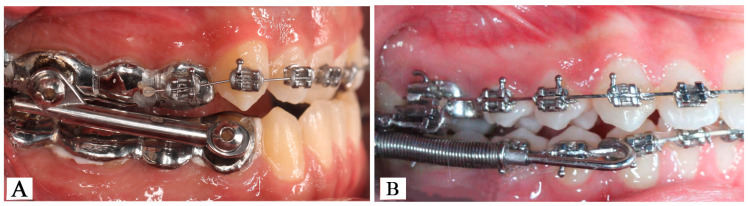
The intraoral application of Herbst (**A**) and Forsus (**B**) appliances.

**Figure 2 jcm-14-00096-f002:**
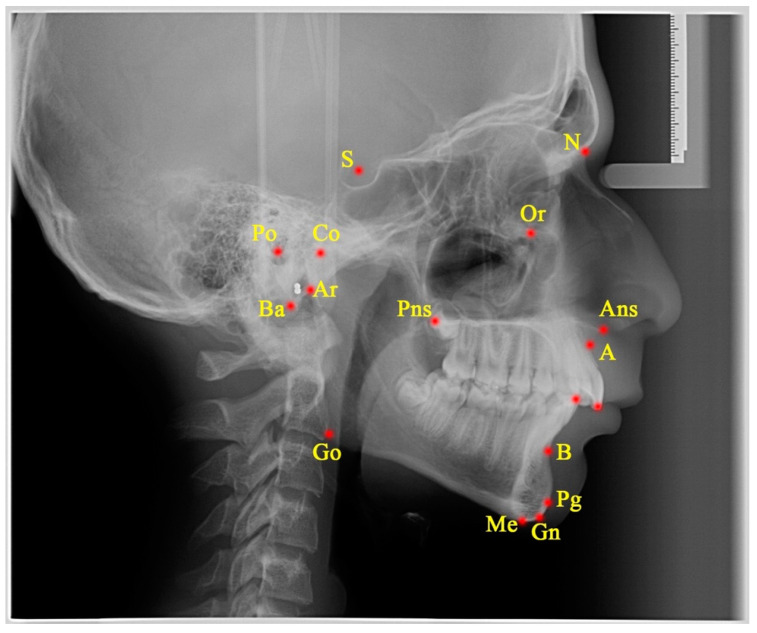
The cephalometric points used in the measurements. Yellow letters marked with red dots indicate skeletal cephalometric landmarks. See Appendix A for abbreviation explanation.

**Figure 3 jcm-14-00096-f003:**
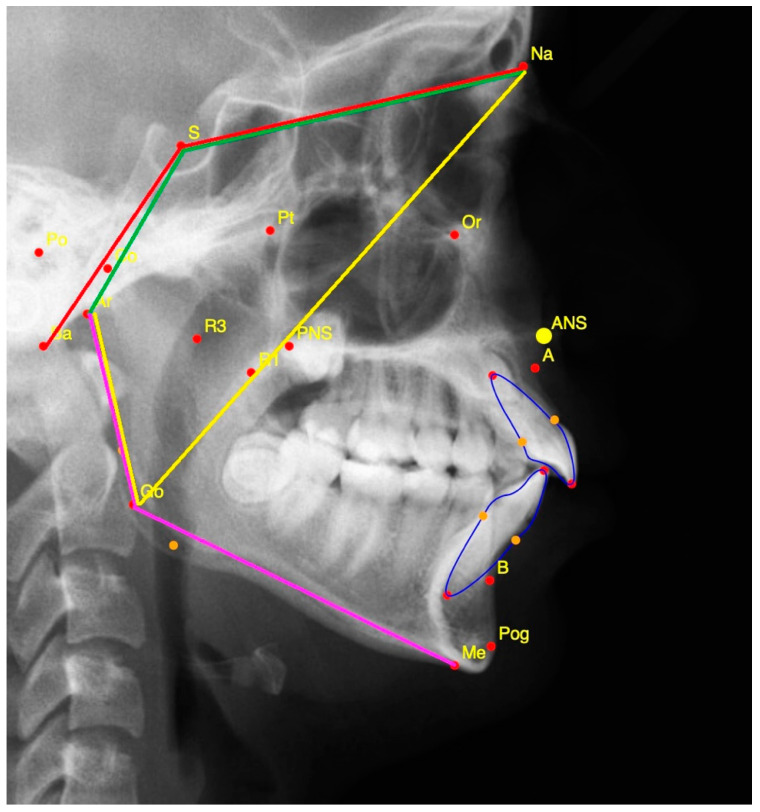
Cranial base measurements: N-S-Ba (shown in red); N-S-Art (shown in green); Ar-Go-Me (Shown in pink); Ar-Go-N (shown in yellow). See Appendix A for abbreviation explanation.

**Figure 4 jcm-14-00096-f004:**
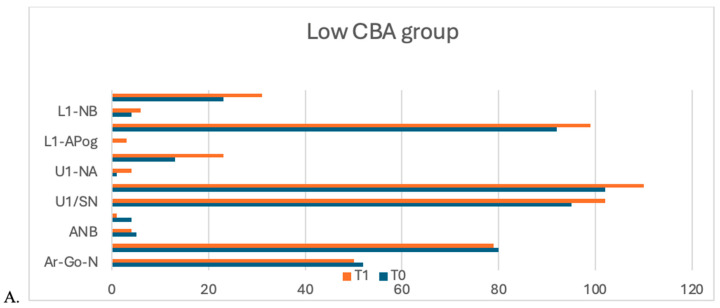
Diagram of the results for the low CBA (**A**), medium CBA (**B**), and high CBA (**C**) groups. CBA: Cranial Base Angle.

**Table 1 jcm-14-00096-t001:** Demographic characteristics.

		Low CBA	Medium CBA	High CBA	*p* Values
Gender	Male	7 (54.0)	4 (30.8)	4 (30.8)	0.102 *
	Female	6 (46.0)	9 (69.2)	9 (69.2)	
CVM Stage	Stage 4	5 (38.0)	3 (23.0)	7 (54.0)	0.273 *
	Stage 5	8 (62.0)	10 (77.0)	6 (46.0)	
Hand–Wrist Stage	Stage 6	4 (30.8)	1 (7.6)	2 (15.4)	0.531 *
	Stage 7	2 (15.4)	2 (15.4)	2 (15.4)	
	Stage 8	1 (7.6)	4 (30.8)	5 (38.4)	
	Stage 9	6 (46.2)	6 (46.2)	4 (30.8)	
Appliance Type	Forsus	7 (54.0)	7 (54.0)	6 (46.0)	0.902 *
	Herbst	6 (46.0)	6 (46.0)	7 (54.0)	
Orthopedic Treatment Duration (year)		0.69 ± 0.08	0.70 ± 0.07	0.70 ± 0.11	0.974 **
Orthopedic and Orthodontic Treatment Duration (year)		2.62 ± 1.29	2.41 ± 0.35	2.15 ± 0.56	0.337 **
Age (year)		14.62 ± 1.12	14.38 ± 0.96	14.08 ± 1.04	0.461 **

CBA: cranial base angle. CVM: Cervical Vertebral Maturation. Categoric data are given as frequency (percentage). Age is given as mean ± standard deviation. * Result of Pearson Chi-square test. ** Result of One-Way ANOVA.

**Table 2 jcm-14-00096-t002:** Comparison of values at the beginning of treatment between groups.

Type	Measurement	Low CBA	Medium CBA	High CBA	*p* Values
**Cranial Base Measurements**	N-S-Ba	128.01 ± 2.02 ^a^	134.64 ± 0.92 ^b^	139.31 ± 2.42 ^c^	**<0.001 ***
	S-Ba	45.92 ± 4.26	44.08 ± 3.75	47.02 ± 3.26	0.130 *
	N-S-Art	122.62 ± 3.34 ^a^	128.24 ± 3.78 ^b^	132.08 ± 4.22 ^c^	**<0.001 ***
	Ar-N	80.6 ± 5.8	82.32 ± 4.92	85.25 ± 4.39	0.078 *
	S-Ar	35.25 ± 3.93	32.72 ± 3.23	34.74 ± 2.29	0.145 *
	S-N	65.85 ± 4.12	65.78 ± 3.19	65.78 ± 3.81	0.999 *
	Ar-Go-Me	119.95 ± 33.75	129.83 ± 7.95	128.4 ± 5.96	0.576 *
	Ar-Go-N	52.5 ± 5.25	54.86 ± 5.43	54.22 ± 5.02	0.521 *
**Sagittal and Vertical Skeletal Measurements**	N-Go-Me	75.82 ± 7.19	74.95 ± 6.26	74.2 ± 4.14	0.778 *
	Go-Me	59.44 ± 5.23	61.65 ± 4.71	61.72 ± 4.62	0.445 *
	SNA	80.21 ± 3.48	80.02 ± 3.14	78.27 ± 2.68	0.264 **
	SNB	74.38 ± 3.20	75.48 ± 2.59	72.96 ± 1.96	0.063 *
	ANB	5.85 ± 2.16	4.54 ± 1.66	5.62 ± 1.26	0.133 *
	Wits	4.65 ± 2.4	3.25 ± 2.44	5.28 ± 2.15	0.100 *
	ANS-Me	61.65 ± 5.36	60.58 ± 6.31	61.72 ± 4.98	0.866 *
	SN/GoGn	34.03 ± 8.18	33.43 ± 5.81	34.76 ± 4.04	0.796 *
	SN/PP	7.97 ± 3.41	9.12 ± 1.9	10.51 ± 3.79	0.232 *
	PP/GoGn	26.06 ± 8.37	24.32 ± 5.53	24.26 ± 4.91	0.790 *
	Na-Me	109.45 ± 6.58	108.56 ± 9.54	110.54 ± 7.29	0.834 *
	S-Go	70.17 ± 7.26	68.39 ± 5.28	69.55 ± 5.74	0.755 *
	S-Go/N-Me	64.16 ± 5.67	63.18 ± 3.97	62.91 ± 2.7	0.780 *
	N-ANS	50.19 ± 4.02	49.82 ± 4.34	51.05 ± 4.45	0.775 *
	ANS-Gn	58.28 ± 5.46	57.07 ± 6.28	58.24 ± 4.88	0.845
	N-Gn	108.5 ± 6.62	107.11 ± 9.56	109.3 ± 7.2	0.810 *
	CoA	83.38 ± 6.17	93.95 ± 28.04	86.83 ± 6.12	0.224 *
	CoGn	108.75 ± 7.39	113.72 ± 7.76	111.16 ± 7.15	0.275 *
	Mx/Md Diff	20.86 ± 2.6	22.5 ± 4.35	20.06 ± 3.35	0.305 *
**Incisor Measurements**	U1/SN	95.73 ± 9.73	97.75 ± 8.41	100 ± 9	0.526 *
	U1/PP	102.19 ± 11.9	106.61 ± 7.48	110.52 ± 8.62	0.146 *
	U1-NA	1.88 ± 3.64	3.55 ± 3.07	4.72 ± 2.84	0.111 *
	U1/NA	13.98 ± 11.61	18.15 ± 8.99	22.2 ± 8.74	0.147 *
	L1-APog	−0.16 ± 3.55	0.25 ± 2.5	−0.12 ± 1.75	0.898 *
	IMPA	92.58 ± 6.64	90.47 ± 6.93	94.36 ± 5.35	0.302 *
	L1-NB	4.48 ± 3.38	4.09 ± 2.57	4.47 ± 2.05	0.910 *
	L1/NB	23.85 ± 9.07	23.01 ± 6.48	24.16 ± 6.17	0.897 *
	Interincisal Angle	108.64 ± 87.6	134.28 ± 13.34	128.19 ± 8.91	0.299 *
	Overbite	4.57 ± 2.5	3.75 ± 1.66	4.45 ± 1.9	0.508 *
	Overjet	6.40 ± 1.98	6.72 ± 2.58	7.82 ± 2.73	0.352 **
**Soft Tissue Measurements**	Nasolabial Angle	111.22 ± 12.93	116.48 ± 8.07	112.12 ± 8.69	0.324 *
	Lower Lip–E Line	0.03 ± 3.31	−0.63 ± 2.39	−0.85 ± 2.35	0.744 *
	Upper Lip–E Line	−1.44 ± 2.54	−1.4 ± 2.12	−1.16 ± 1.36	0.913 *
	Soft Tissue Convexity	125.59 ± 5.16	124.92 ± 3.26	126.95 ± 4.23	0.415 *

Data are given as mean ± standard deviation. CBA: cranial base angle. * Result of One-Way ANOVA test. ** Result of Kruskal–Wallis test. Same letters within lines indicate similarity between groups, while different letters indicate difference between groups. Statistically significant degree is given as *p* < 0.05.

**Table 3 jcm-14-00096-t003:** Comparison of pre- and post-treatment values of cranial base angle groups within the group.

Type	Measurements	Low CBA			Medium CBA			High CBA		
		T0	T1	*p* Values	T0	T1	*p* Values	T0	T1	*p* Values
**Cranial Base Measurements**	N-S-Ba	128.01 ± 2.02	128.98 ± 2.28	0.083 *	134.64 ± 0.92	134.45 ± 2.81	0.764 *	139.31 ± 2.42	137.7 ± 4.61	0.203 *
	SBa	45.92 ± 4.26	45.24 ± 4.66	0.275 *	44.08 ± 3.75	43.87 ± 2.88	0.806 *	47.02 ± 3.26	47.32 ± 3.49	0.726 *
	N-S-Art	122.62 ± 3.34	122.64 ± 3.46	0.973 *	128.24 ± 3.78	128.06 ± 4.25	0.702 *	132.08 ± 4.22	130.97 ± 5.09	0.124 *
	Ar-N	80.6 ± 5.8	80.88 ± 4.59	0.772 *	82.32 ± 4.92	81.89 ± 4.32	0.600 *	85.25 ± 4.39	85.28 ± 4.89	0.946 *
	S-Ar	35.25 ± 3.93	35.13 ± 4.07	0.782 *	32.72 ± 3.23	32.06 ± 2.86	0.179 *	34.74 ± 2.29	34.95 ± 2.2	0.742 *
	S-N	65.85 ± 4.12	66.15 ± 2.87	0.655 *	65.78 ± 3.19	65.78 ± 3.38	0.989 *	65.78 ± 3.81	66.28 ± 4.3	0.170 **
	Ar-Go-Me	119.95 ± 33.75	125.93 ± 9.64	0.224 **	129.83 ± 7.95	130.66 ± 5.38	0.567 *	128.4 ± 5.96	128.98 ± 4.18	0.642 *
	Ar-Go-N	52.5 ± 5.25	50.39 ± 6.1	**0.005 ***	54.86 ± 5.43	55.28 ± 4.43	0.695 *	54.22 ± 5.02	53.16 ± 4.03	0.145 *
**Sagittal and Vertical Skeletal Measurements**	N-Go-Me	75.82 ± 7.19	75.55 ± 6.72	0.663 *	74.95 ± 6.26	75.39 ± 6.61	0.521 *	74.2 ± 4.14	75.82 ± 4.27	0.075 *
	Go-Me	59.44 ± 5.23	60.35 ± 4.72	0.351 *	61.65 ± 4.71	61.22 ± 5.25	0.468 *	61.72 ± 4.62	61.52 ± 3.96	0.846 *
	SNA	80.21 ± 3.48	79.28 ± 3.49	**0.023 ***	80.02 ± 3.14	79.2 ± 3.83	0.152 *	78.27 ± 2.68	78.32 ± 2.50	0.862 *
	SNB	74.38 ± 3.2	74.93 ± 3.31	0.262 *	75.48 ± 2.59	75.84 ± 3.06	0.319 *	72.96 ± 1.96	73.56 ± 2.10	0.468 *
	ANB	5.85 ± 2.16	4.36 ± 2.02	**0.002 ***	4.54 ± 1.66	3.59 ± 1.70	**0.049 ***	5.65 ± 1.23	4.38 ± 1.33	**<0.001 ***
	Wits	4.65 ± 2.4	1.31 ± 1.32	**<0.001 ***	3.25 ± 2.44	0.18 ± 2.42	**<0.001 ***	5.28 ± 2.15	1.56 ± 1.89	**<0.001 ***
	ANS-Me	61.65 ± 5.36	62.85 ± 4.45	0.077 *	60.58 ± 6.31	61.26 ± 7.26	0.197 *	61.72 ± 4.98	64.21 ± 4.1	**0.025 ****
	SN/GoGn	34.03 ± 8.18	34.37 ± 8.79	0.670 *	33.43 ± 5.81	33.56 ± 7.04	0.838 *	34.76 ± 4.04	35.42 ± 6.2	0.784 **
	SN/PP	7.97 ± 3.41	8.28 ± 3.67	0.385 *	9.12 ± 1.9	9.12 ± 2.25	0.989 *	10.51 ± 3.79	10.72 ± 3.64	0.266 **
	PP/GoGn	26.06 ± 8.37	26.11 ± 8.56	0.952 *	24.32 ± 5.53	24.42 ± 6.4	0.871 *	24.26 ± 4.91	24.69 ± 7.3	0.667 *
	N-Me	109.45 ± 6.58	111.63 ± 5.93	0.077 *	108.56 ± 9.54	109.32 ± 10.82	0.351 *	110.54 ± 7.29	114.28 ± 5.24	**0.005 ****
	S-Go	70.17 ± 7.26	71.17 ± 8.59	0.330 *	68.39 ± 5.28	70.09 ± 6.36	**0.021 ***	69.55 ± 5.74	72.12 ± 6.41	**0.033** **
	S-Go/N-Me	64.16 ± 5.67	63.76 ± 6.94	0.566 *	63.18 ± 3.97	64.32 ± 4.77	0.069 *	62.91 ± 2.7	63.14 ± 5.06	0.817 *
	N-ANS	50.19 ± 4.02	50.95 ± 3.78	0.293 *	49.82 ± 4.34	50.08 ± 4.86	0.649 *	51.05 ± 4.45	52.13 ± 4.02	**0.025 ***
	ANS-Gn	58.28 ± 5.46	59.93 ± 4.42	0.057 *	57.07 ± 6.28	58.48 ± 7.19	**0.026 ***	58.24 ± 4.88	60.94 ± 4.4	**0.006 ****
	N-Gn	108.5 ± 6.62	110.89 ± 5.35	0.082 *	107.11 ± 9.56	108.57 ± 11.03	0.102 *	109.3 ± 7.2	113.08 ± 5.14	**0.003 ****
	Co-A	83.38 ± 6.17	83.98 ± 4.76	0.564 *	93.95 ± 28.04	85.65 ± 6.77	0.834 **	86.83 ± 6.12	87.81 ± 5.86	0.270 *
	Co-Gn	108.75 ± 7.39	111.66 ± 6.13	**0.045 ***	113.72 ± 7.76	114.68 ± 9.66	0.162 **	111.16 ± 7.15	114.37 ± 6.32	**0.014 ****
**Incisor Measurements**	U1/SN	95.73 ± 9.73	102.62 ± 5.22	**0.043** *	97.75 ± 8.41	101.32 ± 4.78	0.169 *	100.00 ± 9.00	96.78 ± 5.31	0.168 **
	U1/PP	102.19 ± 11.9	110.86 ± 5.34	**0.039** *	106.61 ± 7.48	110.18 ± 4.64	0.151 *	110.52 ± 8.62	107.17 ± 6.68	0.182 **
	U1-NA	1.88 ± 3.64	4.18 ± 1.27	**0.030 ***	3.55 ± 3.07	4.65 ± 2.8	0.265 *	4.72 ± 2.84	3.3 ± 1.65	0.107 *
	U1/NA	13.98 ± 11.61	23.32 ± 5.45	**0.025 ***	18.15 ± 8.99	22.42 ± 5.44	0.092 *	22.2 ± 8.74	18.59 ± 5.76	0.216 **
	L1-APog	−0.16 ± 3.55	3.25 ± 2.68	**<0.001 ***	0.25 ± 2.5	2.99 ± 2.05	**0.002 ****	−0.12 ± 1.75	2.91 ± 1.77	**<0.001 ***
	IMPA	92.58 ± 6.64	99.64 ± 7.22	**<0.001 ***	90.47 ± 6.93	96.78 ± 7.21	**<0.001 ***	94.36 ± 5.35	101.09 ± 8.72	**0.009 ***
	L1-NB	4.48 ± 3.38	6.97 ± 2.89	**<0.001 ***	4.09 ± 2.57	6.18 ± 2.03	**0.002 ****	4.47 ± 2.05	7.02 ± 1.8	**0.001 ***
	L1/NB	23.85 ± 9.07	31.58 ± 7.29	**<0.001 ***	23.01 ± 6.48	29.68 ± 6.35	**<0.001 ***	24.16 ± 6.17	32.16 ± 7.14	**0.002 ***
	Interincisal Angle	108.64 ± 87.6	120.73 ± 8.19	**0.048 ****	134.28 ± 13.34	124.33 ± 8.25	**0.002 ***	128.19 ± 8.91	124.33 ± 9.51	0.268 *
	Overbite	4.57 ± 2.5	1.78 ± 1.39	**0.002 ****	3.75 ± 1.66	1.92 ± 0.9	**<0.001 ***	4.45 ± 1.9	2.09 ± 0.71	**0.003 ***
	Overjet	6.40 ± 1.98	3.31 ± 0.87	**0.010 ***	6.72 ± 2.58	3.61 ± 1.14	**0.004 ****	7.82 ± 2.73	3.53 ± 0.53	**<0.001 ***
**Soft Tissue Measurements**	Nasolabial Angle	111.22 ± 12.93	112.94 ± 10.17	0.268 *	116.48 ± 8.07	113.91 ± 8.36	0.141 *	112.12 ± 8.69	110.77 ± 9.32	0.172 *
	Lower Lip–E Line	0.03 ± 3.31	0.25 ± 3.23	0.513 *	−0.63 ± 2.39	−0.65 ± 1.95	0.968 *	−0.85 ± 2.35	−0.54 ± 2.38	0.559 *
	Upper Lip–E Line	−1.44 ± 2.54	−2.53 ± 2.72	**0.011 ***	−1.40 ± 2.12	−1.97 ± 2.22	0.136 *	−1.16 ± 1.36	−2.28 ± 1.78	**0.015 ***
	Soft Tissue Convexity	125.59 ± 5.16	126.25 ± 4.3	0.481 *	124.92 ± 3.26	125.94 ± 4.58	0.192 *	126.95 ± 4.23	127 ± 3.85	0.957 *

Data are given mean ± standard deviation. CBA: cranial base angle. T0: pre-treatment. T1: post-treatment. * Result of Paired Samples-*t* test. ** Result of Wilcoxon Signed-Rank test. Statistically significant degree is given as *p* < 0.05.

**Table 4 jcm-14-00096-t004:** Comparison of changes with treatment between groups.

	Measurements	Low CBA	Norm CBA	High CBA	*p* Values
**Cranial Base Measurements**	N-S-Ba	0.97 ± 1.85	−0.19 ± 2.26	−1.61 ± 4.31	0.136 **
	SBa	−0.68 ± 2.13	−0.21 ± 2.98	0.29 ± 2.94	0.637 *
	N-S-Art	0.02 ± 1.63	−0.18 ± 1.63	−1.12 ± 2.43	0.391 *
	Ar-N	0.28 ± 3.37	−0.42 ± 2.83	0.04 ± 2.02	0.836 *
	S-Ar	−0.12 ± 1.57	−0.66 ± 1.67	0.21 ± 2.22	0.511 *
	S-N	0.31 ± 2.42	−0.01 ± 2.01	0.49 ± 1.09	0.740 *
	Ar-Go-Me	5.98 ± 28.33	0.83 ± 5.09	0.58 ± 4.36	0.336 **
	Ar-Go-N	−2.11 ± 2.24	0.42 ± 3.73	−1.05 ± 2.44	0.131 *
**Sagittal and Vertical Skeletal Measurements**	N-Go-Me	−0.27 ± 2.17	0.44 ± 2.39	1.62 ± 3.01	0.214 *
	Go-Me	0.91 ± 3.37	−0.43 ± 2.07	−0.21 ± 3.78	0.494 *
	SNA	−0.92 ± 1.28	−0.82 ± 1.92	0.05 ± 1.25	0.149 *
	SNB	0.55 ± 1.7	0.36 ± 1.25	0.61 ± 1.86	0.908 *
	ANB	−1.49 ± 1.35	−0.95 ± 1.4	−1.12 ± 1.03	0.510 *
	Wits	−3.35 ± 1.88	−3.06 ± 2.3	−3.72 ± 1.26	0.632 *
	ANS-Me	1.19 ± 2.22	0.68 ± 1.78	2.49 ± 3.92	0.519 **
	SN/GoGn	0.34 ± 2.8	0.13 ± 2.26	0.66 ± 3.98	0.896 **
	SN/PP	0.31 ± 1.23	0 ± 1.28	0.22 ± 1.45	0.823 *
	PP/GoGn	0.05 ± 2.71	0.1 ± 2.17	0.43 ± 3.52	0.949 *
	Na-Me	2.18 ± 4.08	0.75 ± 2.8	3.74 ± 4.77	0.164 *
	S-Go	1 ± 3.55	1.7 ± 2.31	2.57 ± 3.86	0.576 *
	S-Go/N-Me	−0.4 ± 2.45	1.14 ± 2.06	0.23 ± 3.52	0.245 *
	N-ANS	0.76 ± 2.49	0.26 ± 2.02	1.08 ± 1.53	0.523 *
	ANS-Gn	1.65 ± 2.81	1.42 ± 2	2.7 ± 3.74	0.842 **
	N-Gn	2.39 ± 4.54	1.46 ± 2.98	3.78 ± 4.86	0.464 **
	Co-A	0.6 ± 3.64	−8.3 ± 28	0.98 ± 3.05	0.431 **
	Co-Gn	2.92 ± 4.7	0.96 ± 2.91	3.21 ± 4.09	0.225 *
	Mx/Md Diff	2.38 ± 2.1	1.98 ± 1.69	2.14 ± 2.26	0.911 **
**Incisor Measurements**	U1/SN	6.88 ± 10.96	3.57 ± 8.8	−3.22 ± 10.64	0.065 *
	U1/PP	8.67 ± 13.46	3.57 ± 8.39	−3.35 ± 9.82	**0.029** **
	U1-NA	2.31 ± 3.38	1.1 ± 3.39	−1.42 ± 2.95	**0.019** *
	U1/NA	9.34 ± 13.12	4.27 ± 8.42	−3.61 ± 10.23	**0.012** **
	L1-APog	3.41 ± 2.4	2.74 ± 1.52	3.03 ± 1.91	0.700 *
	IMPA	7.06 ± 5.31	6.31 ± 4.33	6.73 ± 7.74	0.925 *
	L1-NB	2.48 ± 2.01	2.08 ± 1.51	2.55 ± 2.15	0.766 *
	L1/NB	7.73 ± 6.39	6.68 ± 4.17	8 ± 7.07	0.801 *
	Interincisal Angle	12.09 ± 91.41	−9.95 ± 9.2	−3.86 ± 11.99	0.310 **
	Overbite	−2.79 ± 1.76	−1.83 ± 1.03	−2.36 ± 2.27	0.404 **
	Overjet	−3.09 ± 2.45	−3.12 ± 2.25	−4.29 ± 2.87	0.584 *
**Soft Tissue Measurements**	Nasolabial Angle	1.72 ± 5.35	−2.57 ± 5.87	−1.35 ± 3.34	0.141 *
	Lower Lip–E Line	0.22 ± 1.15	−0.02 ± 1.38	0.31 ± 1.85	0.855 *
	Upper Lip–E Line	−1.09 ± 1.3	−0.57 ± 1.28	−1.12 ± 1.42	0.502 *
	Soft Tissue Convexity	0.65 ± 3.24	1.02 ± 2.67	0.05 ± 3.51	0.740 *

Data are given mean ± standard deviation. CBA: cranial base angle. * Result of One-Way ANOVA test. ** Result of Kruskal–Wallis test. Statistically significant degree is given as *p* < 0.05.

## Data Availability

The datasets used and/or analyzed during the current study are available from the corresponding author on reasonable request.

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
