# Peer review of "Does Cranial Base Angle Make a Difference in the Effectiveness of Functional Orthopedic Treatment? A Retrospective Cohort Study"

_jcm, 2024, doi:10.3390/jcm14010096_

Round 1

Reviewer 1 Report

Comments and Suggestions for Authors

Thank you for submitting the manuscript titled, "Does Cranial Base Angle Make a Difference in the Effectiveness of Functional Orthopedic Treatment?" It is evident that significant effort has been invested in this study, and the subject is highly relevant to the field. Below are some observations and suggestions that might contribute to further strengthening the manuscript.

Abstract

  • The inclusion of brief background information could help establish the relevance of the research. It may also be helpful to explicitly state the study's aim to provide a clearer context for readers.
  • Offering additional details regarding the sample, such as age and gender, would enhance the reader's understanding.
  • Incorporating specific numerical values, including p-values, in the results section could provide additional depth and clarity.

Introduction

  • While the introduction provides a comprehensive background, including a clearer articulation of the rationale behind the study and the specific literature gap it addresses might add to its overall impact.

Materials and Methods

  • The inclusion and exclusion criteria might benefit from being presented in separate paragraphs, which could improve clarity.
  • Further elaboration on the sample characteristics, such as demographic details, systemic conditions, and treatment history, may strengthen the section.
  • If the cervical vertebral maturation (CVM) stage was considered, detailing its role in the methodology could be valuable.
  • The treatment duration and whether it varied among groups might be worth including.
  • The small sample size is acknowledged in the manuscript, but expanding on this aspect and its implications for generalizability could be useful.

Results

  • Although the tables provide detailed data, adding a textual summary of the most significant findings could enhance readability. Highlighting key results throughout this section might make the information more accessible.

Discussion

  • Expanding the discussion by incorporating comparisons with additional relevant studies, where available, could further contextualize the findings.
  • Including a more detailed consideration of the limitations, particularly with respect to the sample size, and suggesting directions for future research might enrich this section.

Conclusion

  • The conclusion effectively summarizes the key findings and does not appear to require significant changes.

These observations are offered in the hope of supporting the development of the manuscript into its best possible form. 

Best regards!

Author Response

Dear Reviewer, 

We have carefully reviewed your comments and fully agree with your suggestions. Thank you for thoroughly examining our manuscript and guiding us in improving it further. Our responses to your comments and the updated version of the manuscript are attached.
Best regards

Reviewer 2 Report

Comments and Suggestions for Authors

Dear Authors,

Congratulations on the nice manuscript. You clearly put a lot of energy and effort into this manuscript and that shows!

There are a few things that would elevate your already
respectable manuscript to an even higher level in my opinion. Here are my respectful recommendations:

First of all:
Please include some illustrations. For example, one cephalometric picture showing the points, then another one showing the angles or as needed to be understandable. So please illustrate supplementary tables 1 and 2.
Please put the key findings such as “the High CBA group showed increases in ANS-Me, N-Me, N-ANS, and N-Gn lengths” into illustrations as well. I would for example make 3 illustrations 1 for the low, medium and high cba group to visually illustrate on them the key changes and differences. This would make it much easier to visualize the differences.
Please minimally expand the section about the theories and evidence the effects of removable functional appliances to tie the finding more to practical clinical relevance.
Please describe the treatment protocols wit Herbst and Forsus in more details. More detailed an all-encompassing description of the protocol here would greatly enhance the possible repeatability of the study and the possible further inclusion in future meta reviews etc.
 Please consider adding pictures here too about the devices and the detailed way they were applied. What were the selection criteria and exclusion criteria for the patient treated with the fixed functional appliances? This effects the sample. How long were the therapies: changes observed might also depend on the length of treatment.
Your findings are most interesting and very relevant, but they should be briefly illuminated from a modern, contemporary perspective. Your article describes important differences which imply changes to the face as well. This is of importance and the facial implications of your findings should be discussed at least in a paragraph. Describe your key findings in how it related to the proportions of the face and the shape of the head. With modern soft-tissue focus orthodontics (not denying the importance of hard tissues of course!) the importance of the face and aesthetic is key.
One article I would recommend you consider regarding a modern take the face, aesthetics, it’s structure is this new MDPI paper: The potential of AI-powered face enhancement technologies in face-driven orthodontic treatment planning APPLIED SCIENCES-BASEL 14 : 17 Paper: 7837 , 21 p. (2024) You may find it interesting and helpful for your article and I hope it helps highlight thought inducing aspects.
It seems to me as though there is a change in the text format between line 171 and 172, please check if it just the file and reformat if necessary.

Thank you for your hard work. The manuscript presents an interesting and timely contribution to the field. It is and interesting and  Authors have clearly invested significant effort in designing the study and collecting data, which reflects their dedication to advancing knowledge in this area. There is considerable potential for this research to make a significant impact once further refinements are made.

I appreciate the authors' thoughtful approach and encourage them to consider and implement my respectful recommendations. All the best for your continued work!

Author Response

(The authors gave the same response as above.)

Round 2

Reviewer 2 Report

Comments and Suggestions for Authors

Dear Authors,
Thank you for your hard work, I think your manuscript is a very valuable addition to the orthodontic scientific landscape.
It was a pleasure to read the new version and I think the improvements you made elevated your already very nice article to it's peak potential.

Author Response

Dear Reviewer,  
Thank you for your valuable comments.